# Mild Hydrothermal Treatment for Improving Outturn of Basmati Rice

**D. M. C. Champathi Gunathilake [1,\*]** and **Wijitha Senadeera [2]**

1 Department of Food Technology, Institute of Agro-Technology and Rural Sciences, University of Colombo, Hambantota 82004, Sri Lanka
2 School of Engineering, University of Southern Queensland, Springfield 4300, Australia
\* Correspondence: champathi@uciars.cmb.ac.lk

**Abstract:** Hydrothermal treatment of rice, called "Parboiling", is an ancient traditional process in Asian countries. It consists of soaking rough rice in water and steaming it, and it both reduces the level of grain breakage and increases head yield of rice during milling. However, parboiling of rice is associated with some drawbacks regarding consumer preferences: the loss of its aroma, reduced rice-kernel whiteness and increased kernel hardness. This research study was carried out to develop a mild hydrothermal treatment that could be applied to basmati paddy by controlling hydrothermal treatment, i.e., soaking water temperature, steaming pressure and time. The Basmati 370 paddy variety was used for this study. The results revealed that, by soaking the paddy in hot water ($70 \pm 2\ ^\circ$C) for 75 min and steaming the soaked paddy for 20 min with non-pressurized steam at atmospheric pressure, and soaking the paddy for 120 min in hot water ($70 \pm 2\ ^\circ$C) and steaming the soaked paddy for 4 min with pressurized steam ($4\ \text{kg/cm}^2$), the optimum treatments are achieved. These optimum hydrothermal treatments were able to produce high head rice yield and preserve the basmati aroma, colour, hardness and palatability characteristics similar to non-parboiled basmati rice. Further, nutritional values such as vitamin B and protein content were also significantly preserved by these mild hydrothermal treatments. These optimized treatment combinations achieved minimized grain breakage while increasing head rice yield during milling and, at the same time, preserved basmati aroma, kernel whiteness, cooking and palatability characteristics similar to non-parboiled rice.

**Keywords:** basmati paddy; hydrothermal treatment; parboiling; soaking; steaming; quality of rice

## 1. Introduction

Rice (*Oryza Sativa*) (the raw crop call is called paddy) is one of the most important and extensively grown food crops in the world. It is the staple food of half of the world's population. Rice is mainly produced and consumed in the Asian region. The major Indian aromatic rice variety is called "Basmati", which is famous in the world. India is also the largest producer and exporter of basmati rice [1].

Hydrothermal treatment of rice, commonly called "parboiling of rice" is a traditional process in Southern Asia since ancient times. It has been reported that hydrothermally treated (parboiled) rice is also predominant in South Asian countries such as India, Sri Lanka and Bangladesh. Generally, the hydrothermal treatment process consists of three main stages: soaking the cleaned raw rough rice to saturation moisture content, gelatinization of rice starch by adding heat to the moist kernels through steaming and drying the product to a moisture content suitable for milling or storage. Water and heat are the two main elements in the hydrothermal process, which is the main process of parboiling. It can be achieved through a variety of methods that differ basically in the intensity of the heat of hydrothermal treatment [2–5]. Parboiling of rice causes physical and chemical modifications in the grain, leading to favourable changes such as easier shelling, higher

head rice yield, fewer broken grains, increased resistance to insects, firmer cooked rice texture, less solids loss during cooking and washing, better retention of nutrients (e.g., proteins, vitamins and minerals) and higher oil content in the bran [2–5].

Although parboiled rice has been reported have many advantages, parboiling of basmati paddy is not popular, because of parboiling of basmati paddy might cause a loss of aromatic value, which is unique for basmati rice, and other organoleptic qualities such as taste, colour, aroma, texture and other palatability characteristics. Many previous studies reported the severity of hydro and thermal (degree of hydrothermal) treatment caused a loss of aromatic volatile compounds from the rice [6,7]. Generally, basmati rice has reported to have a high broken grain percentage (BGP) compared to other varieties during milling due to its kernel having a long, slender shape. Hence, this study was carried out to develop mild hydrothermal treatment for basmati paddy in order to increase head rice yield (HRY) while keeping its aroma and kernel whiteness (KW). Previous studies revealed that parboiled rice quality mainly depends on the amount of water absorbed during soaking and steaming duration. The level (high or low) of soaking-water temperature was only affected water absorbing (hydration) time i.e., hot-water soaking absorbed quickly, and cold-water soaking absorbed slowly into the rice kernel. Parboiled rice quality mainly depends on the absorbed water amount in the rice kernel. Therefore, in this research study, two temperature ranges were used for soaking, i.e., hot water at 70 °C and water at ambient temperature (water at ambient room temperature is used for soaking paddy in traditional parboiling) [2,5,8,9]. As control treatments, full soaking and steaming time were given to the two basmati samples. Further, this study aims to overcome drawbacks of the hydrothermal process for basmati paddy by controlling soaking (two temperatures and several different times under these two temperatures) and steaming (two pressures and several different times under these two pressures), which would minimize grain breakage during milling and preserve aroma and whiteness of the milled rice. At the same time, mild hydrothermal treatment was required to preserve the organoleptic qualities, i.e., basmati aroma, texture, taste and other palatability characteristics of basmati rice.

Hence, the main objective of this study was to develop optimized mild hydrothermal treatments for basmati rice that were able to produce palatability characteristics similar to non-treated raw basmati rice while maximizing head rice yield kernel whiteness. At the same time, this treatment aimed to preserve aroma, texture and palatability characteristics of raw basmati rice.

## 2. Materials and Methods

Basmati 370, a high yield variety which is the most popular among Punjab farmers, was used for this experiment. Mild hydrothermal treatments were carried out by soaking in different time and temperature combinations followed by steaming in different time and pressure combinations. Two temperature ranges, hot water at $70 \pm 2$ °C and water at room temperature ($28 \pm 3$ °C), were selected as soaking treatments. Basmati rice has intermediate amylose content. Therefore, its gelatinization temperature is 72 °C, hence the hot water temperature needs to be kept below gelatinization temperature to absorb water into the rice kernel without the occurrence of gelatinization [8]. Therefore, this study maintained the hot water temperature at 70 °C. Hot water soaking and ambient-temperature soaking were carried out for 4 different durations. The paddy samples soaked for 2 different temperatures and 8 different time combinations were then steamed with two pressure combinations of pressurized steam at 4 kg/cm$^2$ and non-pressurized steam at atmospheric pressure (1.0333 kg/cm$^2$) for 6 different time combinations. Table 1 shows the treatment combinations. After soaking and steaming treatments, paddy samples underwent drying. All paddy samples were dried in a laboratory electric dryer until they reached a moisture content of 14% (wet basis). When the paddy moisture content reached 16% (wet basis) during drying, a three-hour tempering period was applied by placing the paddy samples in fully covered bins in order to prevent crack formation in the rice kernels. Different hydrothermal-treatment effects were evaluated for kernel whiteness, broken grain

percentage, head rice yield percentage kernel hardness and sensory (organoleptic) qualities. Paddy samples were processed according to the method outlined by Bal [9], in which 250 g of dried paddy (14% moisture content) samples were de-hulled using the Satake, Japan laboratory testing husker and the brown (unpolished) rice grains were polished for 95 s by the Satake laboratory rice mill (abrasive type polisher). Raw paddy (non-treated) and full hydrothermally treated paddy samples that were obtained with modern parboiling techniques (paddy soaked in hot water at $70 \pm 2\,^{\circ}$C for 4.5 h and steamed 10 min with pressurized steam at $4\,\text{kg/cm}^2$ pressure) were milled at the same degree and were used as control treatment for the comparison of quality and palatability characteristics. For comparison purposes, the rice sample was prepared with the traditional method of hydrothermal treatment, namely, the paddy was soaked in water at ambient temperature for 48 h followed by 10 min pressurized (at $4\,\text{kg/cm}^2$ pressure) steaming.

**Table 1.** Soaking and steaming treatment combinations.

| | Soaking Time (min) | Steaming Time (min) | | Soaking Time (min) | Steaming Time (min) |
|---|---|---|---|---|---|
| Hot water at $70 \pm 2\,^{\circ}$C soaking and pressurized steaming at $4\,\text{kg/cm}^2$ pressure | 15 | 1.0 | Hot water at $70 \pm 2\,^{\circ}$C soaking and non-pressurized water $(1.0333\,\text{kg/cm}^2)$ steaming | 15 | 10 |
| | 15 | 2.5 | | 15 | 20 |
| | 15 | 4.0 | | 15 | 30 |
| | 45 | 1.0 | | 45 | 10 |
| | 45 | 2.5 | | 45 | 20 |
| | 45 | 4.0 | | 45 | 30 |
| | 75 | 1.0 | | 75 | 10 |
| | 75 | 2.5 | | 75 | 20 |
| | 75 | 4.0 | | 75 | 30 |
| | 120 | 1.0 | | 120 | 10 |
| | 120 | 2.5 | | 120 | 20 |
| | 120 | 4.0 | | 120 | 30 |
| Water at ambient temperature $(28 \pm 3\,^{\circ}$C) soaking and pressurized steaming at $4\,\text{kg/cm}^2$ pressure | 60 | 1.0 | Water at ambient temperature $(28 \pm 3\,^{\circ}$C) soaking and non-pressurized water $(1.0333\,\text{kg/cm}^2)$ steaming | 60 | 10 |
| | 60 | 2.5 | | 60 | 20 |
| | 60 | 4.0 | | 60 | 30 |
| | 270 | 1.0 | | 270 | 10 |
| | 270 | 2.5 | | 270 | 20 |
| | 270 | 4.0 | | 270 | 30 |
| | 510 | 1.0 | | 510 | 10 |
| | 510 | 2.5 | | 510 | 20 |
| | 510 | 4.0 | | 510 | 30 |
| | 720 | 1.0 | | 720 | 10 |
| | 720 | 2.5 | | 720 | 20 |
| | 720 | 4.0 | | 720 | 30 |

*2.1. Measurement of Broken Grain % and Head Rice Yield %*

Broken grains and head rice yield of the treated samples were determined by using a representative working sample of milled rice, of which 100 g was obtained by using a precision electrical sample divider. Grain particles which are smaller than $^3/_4$ of the grain were considered as broken grains and were separated by hand picking, and broken grain percentage was calculated. Accordingly, head rice yield was calculated. These milling characteristics were calculated using the method outlined by Bal [9].

*2.2. Measurement of Rice-Kernel Whiteness*

Rice-kernel whiteness was measured by using a Mini-scan XE plus Hunter Lab Colorimeter, Virginia, USA, model No 45/0-L. In Hunter scale, L measures lightness (whiteness or darkness). The 'L' value varies from 100 for perfect white and zero (0) for perfect black. Hence, L value (lightness value) was used as the whiteness value of the rice kernel.

### 2.3. Measurement of Rice-Kernel Hardness

Compression tests were carried out to measure rice-kernel hardness (yield stress). The Instron, USA, XT2 texture analyser has been adapted to perform a compression test. The textural analyser, having automatic loading rate and chart-plotting facility, was used to determine rice hardness. Force at rupture was considered as the seed hardness. Three replicates were considered for each treatment and the measured forces were averaged.

### 2.4. Procedure for Selection of Optimum Treatment

The optimum mild hydrothermally treated (that best preserved basmati aroma and texture kernel whiteness and obtained highest head rice yield) rice that has similar appearance to non-treated raw rice was selected by two phases of procedures. In the first phase, four optimized treatments were selected based on the following four main treatment combinations (according to Table 2):

1. Hot water ($70 \pm 2\,^\circ$C) soaking and pressurized ($4\,\text{kg/cm}^2$) steaming time combinations;
2. Hot water ($70 \pm 2\,^\circ$C) soaking and non-pressurized ($1.0333\,\text{kg/cm}^2$) steaming time combinations;
3. Ambient-temperature ($28 \pm 3\,^\circ$C) water soaking and pressurized ($4\,\text{kg/cm}^2$) steaming time combinations;
4. Ambient-temperature ($28 \pm 3\,^\circ$C) water soaking and non-pressurized ($1.0333\,\text{kg/cm}^2$) steaming time combinations.

**Table 2.** Selection of optimum treatment from 75 min hot water soaking and 20 min non-pressurized steaming.

| Predicted Optimized Treatment | | | | | |
|---|---|---|---|---|---|
| | | | | **Optimization** | |
| **Process Parameters** | **Target** | **Experiment Range** | | **Importance** | **Optimum Conditions** | **Desirability** |
| Hot water soaking | is in range | 15 min | 120 min | 3 | 62.52 min | |
| Non-pressurized steaming | is in range | 10 min | 30 min | 3 | 25.52 min | |
| **Response** | | | | | **Predicted values** | 0.973 |
| Kernel whiteness (L value) | 62.5 | 60.72 | 69.43 | 4 | 62.718 | |
| Broken grain % | minimize | 7.62 | 25.54 | 5 | 8.396 | |
| Head rice yield % | maximize | 49.94 | 64.83 | 5 | 64.499 | |
| Kernel hardness (kg) | 150.0 | 106.69 | 159.87 | 4 | 150.00 | |
| **Selected optimized treatment** | | | | | | |
| Treatment | Whiteness L value | Broken grain % | | Head rice yield % | Kernel hardness kg | |
| 75 min hot water soaking and 20 min non-pressurized steaming | 64.583 | 7.863 | | 64.520 | 151.449 | |

Surface response methodology by superimposition technique was adopted and the optimum treatment was selected based on the maximum possible head rice yield percentage, highest kernel whiteness, lowest kernel hardness and lowest possible broken grain percentage. The contour plots were prepared for the fitted model and treatments (independent variables) were drawn according to the expected maximum responses (dependent variables) with the aid of Design Expert® DX7 1.6. version.

In the second phase, one optimized mild hydrothermal treatment, for which the treated rice has similar quality to non-treated rice, was selected among the 4 optimum treatments based on preservation of basmati aroma and other organoleptic qualities of cooked rice such as colour, taste and texture. Sensory evaluation was carried out for this selection. Four optimum mild hydrothermal treated rice samples selected in the 1st phase were compared

with control treatments, i.e., non-treated and full hydrothermally treated rice judged by sensory evaluation. DMRT multiple mean comparison techniques were adopted to perform this comparison.

### 2.5. Sensory Evaluation Procedure

Selected samples were cooked: 30g of head rice was immersed in 300 mL of water in glass tube in a boiling water bath for 22 min. After cooking, the contents of the tube were emptied into a perforated circular strainer and kept for 2 min to drain excess water and then served for the sensory test. The cooked rice samples belong to seven different treatments (four optimized mild hydrothermally treated rice samples with non-treated rice and two full hydrothermally treated rice samples) were coded with random numbers and served to 15 trained sensory panelists. Sensory panel members were requested to comment on the colour, odour, taste, texture and overall acceptability, and they were given a structured questionnaire which included a 9-point hedonic scale (ranging from extreme dislike on point 1 and extreme like on point 9). Data of sensory evaluation were analysed by the Friedman test with the aid of the Minitab® computer package. The sums of ranks achieved by the Friedman test for each treatment were compared by the DMRT multiple comparison technique at $\alpha = 0.05$.

## 3. Results

### 3.1. Hydrothermal Effect on Rice-Kernel Whiteness

It was clear from the results that the highest rice-kernel whiteness was observed in non-treated raw basmati rice, and it was 70.77 (L value). However, the lowest whiteness (L value) was reported from full hydrothermally treated rice, which generally followed the traditional method, and was 53.4. The rice-kernel whiteness value of mildly hydrothermally treated rice ranged in between these two values. Increases in hydrothermal treatment time (soaking and steaming duration) cause reduced rice-kernel whiteness. Mild hydrothermal treatment preserved the kernel whiteness in comparison to full hydrothermal treatment. Out of all mild hydrothermal treatments, the minimum whiteness value of 58.29 was reported from the treatment combination of 8.5 h ambient-temperature water soaking and 30 min non-pressurized steaming. This treatment caused a reduction of 16.22 percent of kernel whiteness in comparison to non-treated rice.

### 3.2. Hydrothermal Effect on Broken Grain Percentage and Head Rice Yield Percentage

The results indicate that the level of grain breakage during milling mainly depends on the degree of hydrothermal treatment, i.e., soaking and steaming durations. The lowest broken grain percentage was reported from the samples subjected to full hydrothermal treatment and the highest was reported from non-treated raw rice samples. Increasing soaking and steaming time has directly reduced broken grain percentage. Reduction of broken grain tends to increase head rice yield percentage. Accordingly, full hydrothermally treated rice was observed to achieve high head rice yield.

### 3.3. Hydrothermal Effect on Rice-Kernel Hardness

The results show that rice-kernel hardness increased with increased duration of hydrothermal treatment, i.e., soaking and steaming time. The maximum kernel hardness was observed in samples subjected to full hydrothermal treatment whereas the lowest was reported from non-treated (raw) rice samples. According to the degree of hydrothermal treatment, rice-kernel hardness was altered. Results revealed that mild hydrothermal treatment did not cause hard rice texture.

### 3.4. Selection of Optimum Treatments

The graphical multi-response surface optimization technique by Design Expert software was employed to determine the workable optimum conditions and select optimum treatment (independent variable) in terms of high kernel whiteness, less hardness, less

broken grain percentage and high head rice yield (responses for treatments). The superimposed contour plots, having a common superimposed area for all the expected response for mild hydrothermal treatments, were drawn in order to arrive at the optimum ranges of process parameters.

### 3.4.1. Selection of Optimum Treatment from Hot Water Soaking and Pressurized Steaming Treatment Combinations

Figure 1 shows the superimposed contour plots having common superimposed area for all the response for hot water soaking and pressurized steaming duration treatments. Red dots in Figure 1 show the treatment combinations. The figure indicates that 120 min hot water soaking and 4 min pressurized steaming treatment was in the optimum range, indicated as the yellow-coloured area (Figure 1). Hence, this treatment can be selected as the optimum treatment that reported high kernel whiteness, less hardness, less broken grain percentage and high head rice yield compared to other treatment combinations in this range.

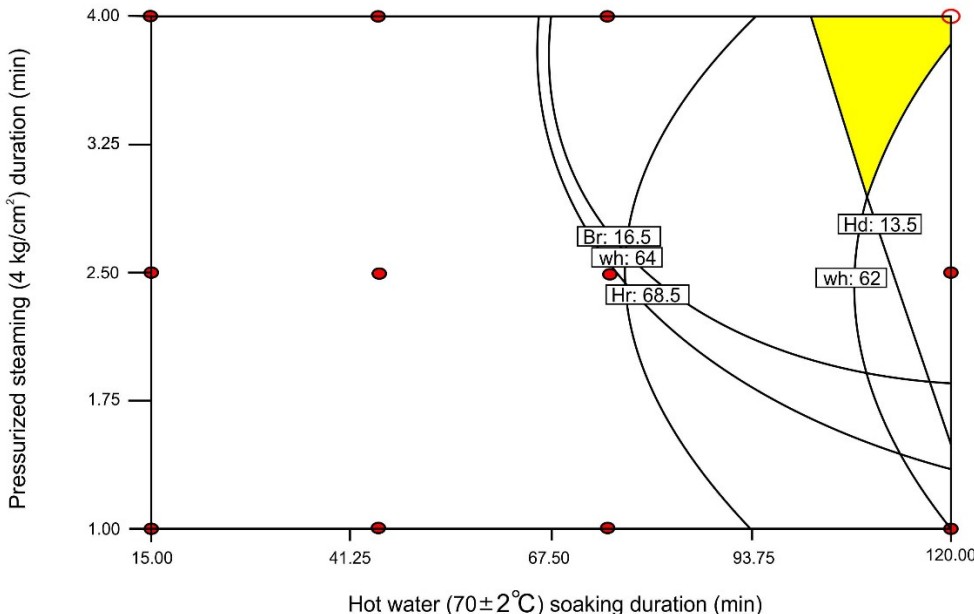

**Figure 1.** Overlaid contour plot for response of hot water soaking and pressurized steaming durations treatment combinations. Explanation: Red dots—treatment combinations; Yellow coloured area—Optimum range.

### 3.4.2. Selection of Optimum Treatment from Hot Water Soaking and Non-Pressurized Steaming Treatment Combinations

Figure 2 shows the superimposed contour plots, having common superimposed area, for all the responses for hot water soaking and non-pressurized steaming duration treatments are in red dots in Figure 2 show the treatment combinations. The rice was subjected to 75 min hot water soaking and 20 min non-pressurized steaming was positioned in the optimum range, indicated in yellow-coloured area (Figure 2). Hence, this treatment can be selected as the optimum treatment that reported high kernel whiteness, less hardness, less broken grain percentage and high head rice yield compared to other treatment combinations in this range.

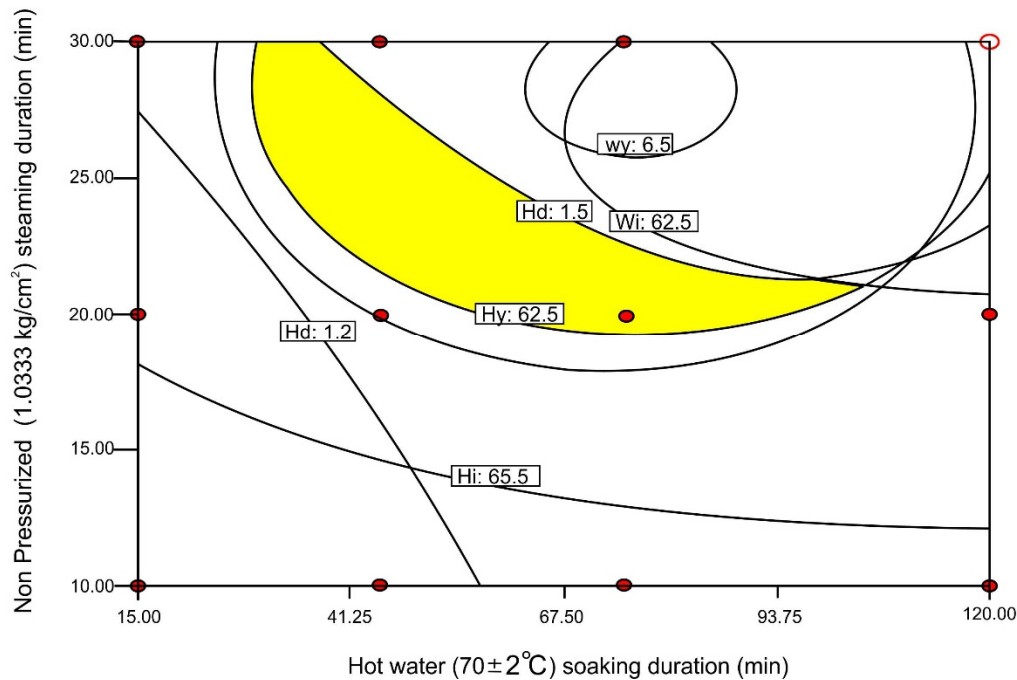

**Figure 2.** Overlaid contour plot for response of hot water soaking and non-pressurized steaming durations treatment combinations. Explanation: Red dots—treatment combinations; Yellow coloured area—Optimum range.

3.4.3. Selection of Optimum Treatment from Ambient-Temperature Water Soaking and Pressurized Steaming Treatment Combinations

Figure 3 shows the superimposed contour plots, having common superimposed area, for all the response for ambient-temperature, (i.e., room-temperature) water soaking and pressurized steaming duration treatments. The rice subjected to 12 h room-temperature water soaking and 4 min pressurized steaming was in the optimum contour range, shown in the figure (Figure 3). Hence, this treatment can be selected as the optimum treatment that reported high kernel whiteness, less hardness, less broken grain percentage and high head rice yield compared to other treatment combinations in this range.

3.4.4. Selection of Optimum Treatment from Ambient-Temperature Water Soaking and Non-Pressurized Steaming Treatment Combinations

Figure 4 shows the superimposed contour plots, having common superimposed area, for all the responses for ambient-temperature water soaking and non-pressurized steaming treatments. Red dots in Figure 4 show the treatment combinations. The rice subjected to 8.5 h normal water soaking and 10 min non-pressurized steaming has been positioned in the optimum range, depicted as the yellow-coloured area (Figure 4). Hence, this treatment can be selected as the optimum treatment that reported high kernel whiteness, less hardness, less broken grain percentage and high head rice yield compared to other treatment combinations in this range.

As a part of the optimization process, the desirability contour plot for treatments was drawn according to the optimized conditions of the response, i.e., kernel whiteness and hardness values, with targets 62.5 and 150.0 kg (as non-treated raw rice) respectively, minimum broken grain percentage and maximum head rice yield percentage (as full hydrothermally treated rice). Maximum equal importance was given to treatments/independent variable (soaking and steaming) when the data were fed to the software. However, the importance given to the responses (dependent variable) were based on their relative contribution to quality of rice. Accordingly, maximum importance was given to broken grain percentage, head rice yield percentage, kernel whiteness and hardness. As a result, the software has predicted optimized values and drawn the contour plot. Predicted optimized

treatment and its values of response, desirability and two selected optimum treatments are given in Tables 2 and 3 and Figures 5 and 6. Consequently, hot water soaking of 75 min and 20 min non-pressurized steaming and 120 min hot water soaking, and 4 min of pressurized steaming treatments were selected as the optimized treatment.

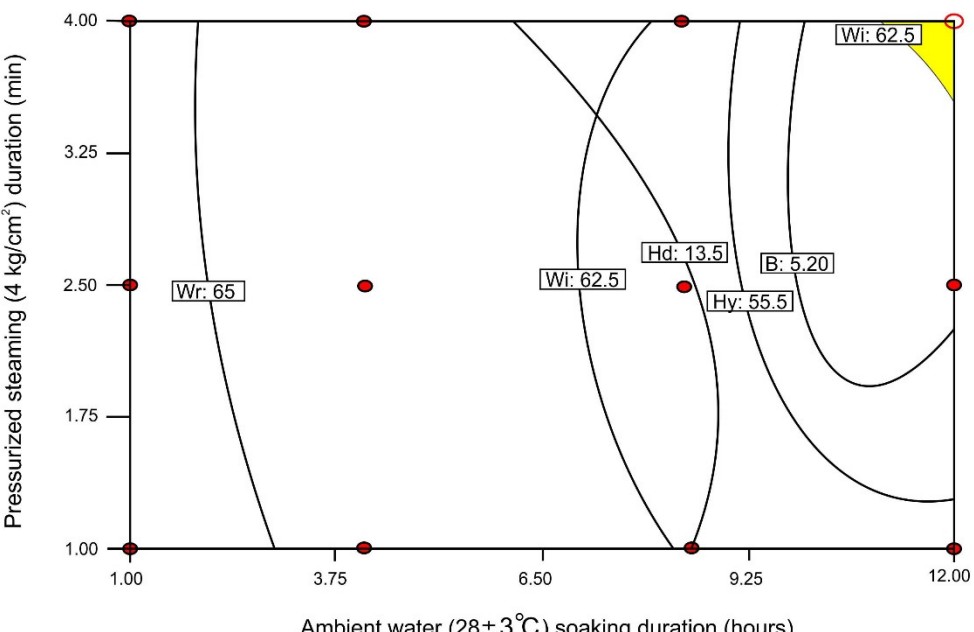

**Figure 3.** Overlaid contour plot for response of ambient-temperature water soaking and pressurized steaming durations treatment combinations. Explanation: Red dots—treatment combinations; Yellow coloured area—Optimum range.

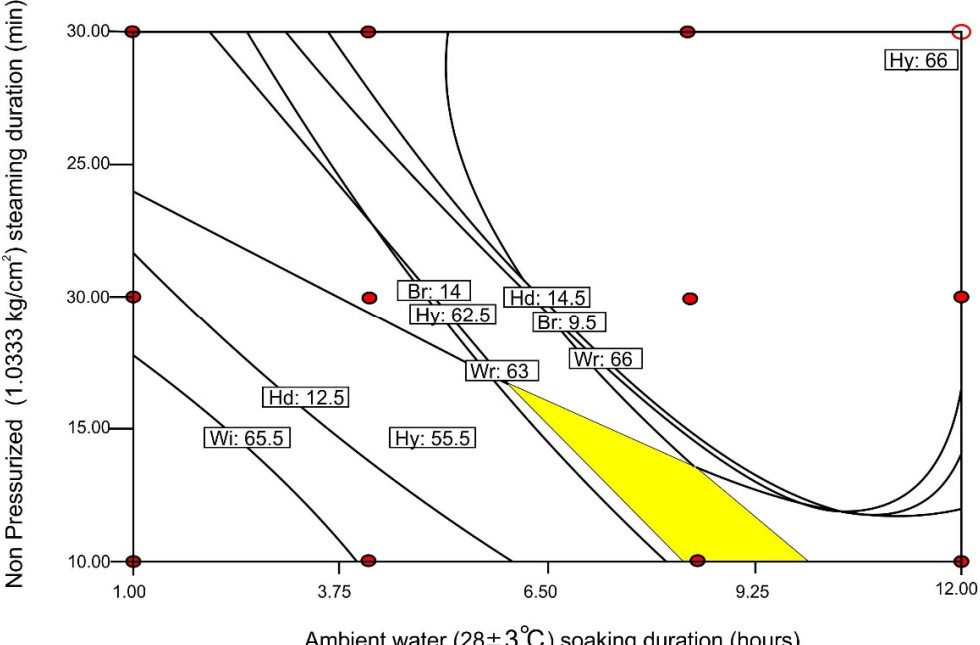

**Figure 4.** Overlaid contour plot for response of ambient-temperature water soaking and non-pressurized steaming durations treatment combinations. Explanation: Red dots—treatment combinations; Yellow coloured area—Optimum range.

**Table 3.** Selection of optimum treatment from 120 min hot water soaking and 4 min pressurized steaming.

| Predicted Optimized Treatment | | | | | | |
|---|---|---|---|---|---|---|
| Process Parameters | Target | Experiment Range | | Importance | Optimization | |
| | | | | | Optimum Conditions | Desirability |
| Hot water soaking | is in range | 15 min | 120 min | 3 | 116.23 min | |
| Pressurized steaming | is in range | 1.0 min | 4.0 min | 3 | 4.00 min | |
| **Response** | | | | | **Predicted values** | 0.983 |
| Kernel whiteness (L value) | 62.5 | 60.43 | 68.23 | 4 | 62.49 | |
| Broken grain % | minimize | 11.31 | 30.09 | 5 | 12.16 | |
| Head rice yield % | maximize | 46.51 | 63.40 | 5 | 63.08 | |
| Kernel Hardness (kg) | 150.0 | 81.12 | 151.22 | 4 | 151.00 | |
| Selected optimized treatment | | | | | | |
| Treatment | Whiteness L value | Broken grain % | | Head rice yield % | Kernel hardness kg | |
| 120 min hot water soaking and 4 min pressurized steaming | 63.19 | 11.31 | | 63.40 | 151.22 | |

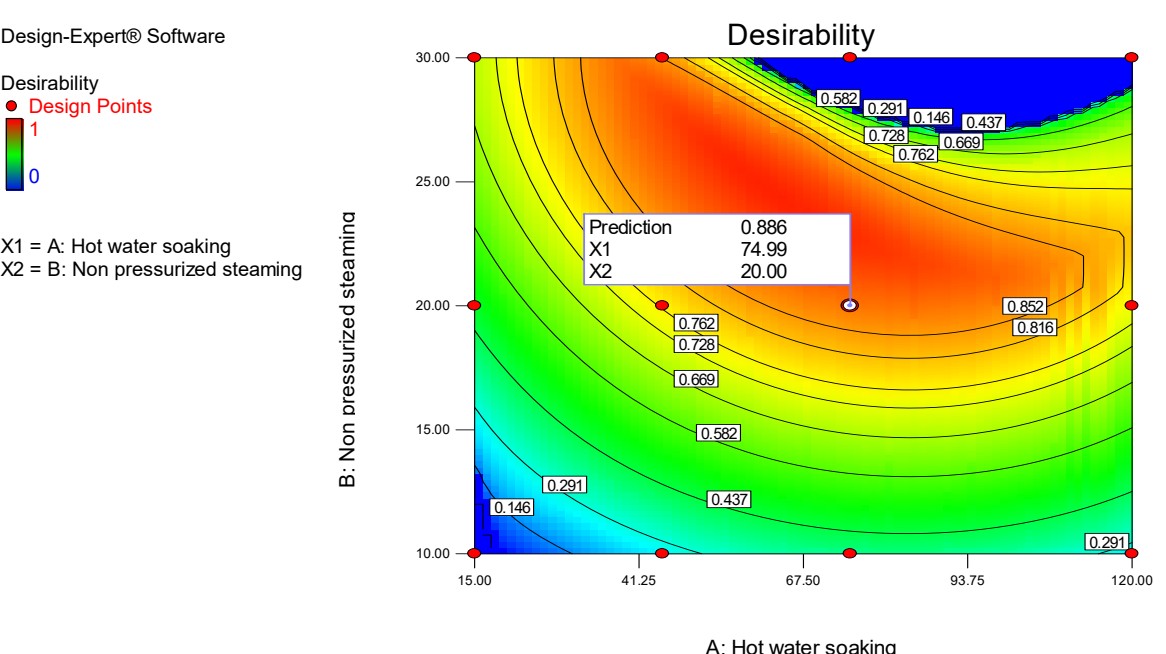

**Figure 5.** Desirability contour plot for optimized treatment: 75 min hot water soaking and 20 min non-pressurized steaming durations.

### 3.5. Selection of the Optimized Mild Hydrothermal Treatment by Sensory Analysis

It is required to select one optimized mild hydrothermal treatment in terms of palatability characteristics, similar to non-treated raw rice, from four optimum treatment combinations selected above. The selected optimum treatment combinations have been selected according to higher kernel whiteness, less hardness, less broken grain percentage and higher head rice yield. These four optimum treatment combinations were evaluated by sensory techniques for identification of the best treatment combination that can produce basmati rice similar to non-treated raw rice in terms of organoleptic qualities.

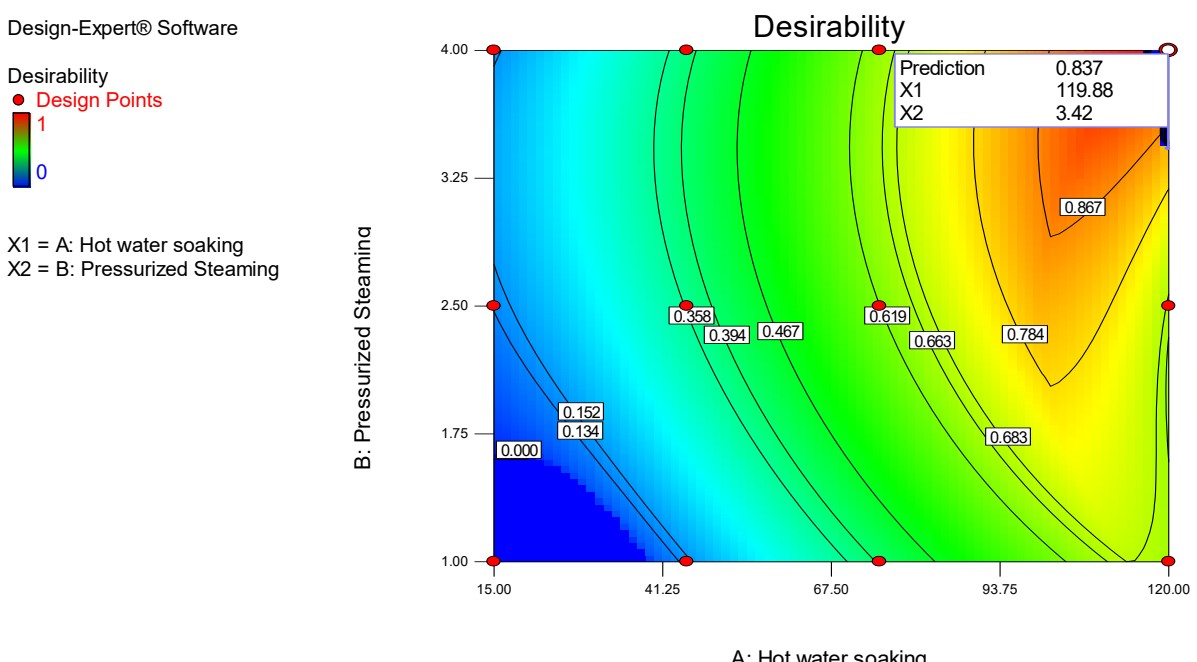

**Figure 6.** Desirability contour plot for optimized treatment: 120 min hot water soaking and 4 min pressurized steaming durations.

*3.6. Comparison of the Organoleptic Properties of Optimized Mild Hydrothermally Treated Rice with Non-Treated and Full Hydrothermally Treated Rice*

Organoleptic properties, namely colour, basmati aroma/odour, taste and texture of mild hydrothermally treated rice, were compared with non-treated and full hydrothermally treated rice with the sensory method. Sensory panelists were requested to comment their overall view (their acceptability) for different treated samples during the sensory test. Figure 7 shows the results of the sum of ranks of Friedman tests for cooked rice colour, basmati aroma/odour, taste, texture and overall acceptability in different hydrothermally treated and non-treated rice samples.

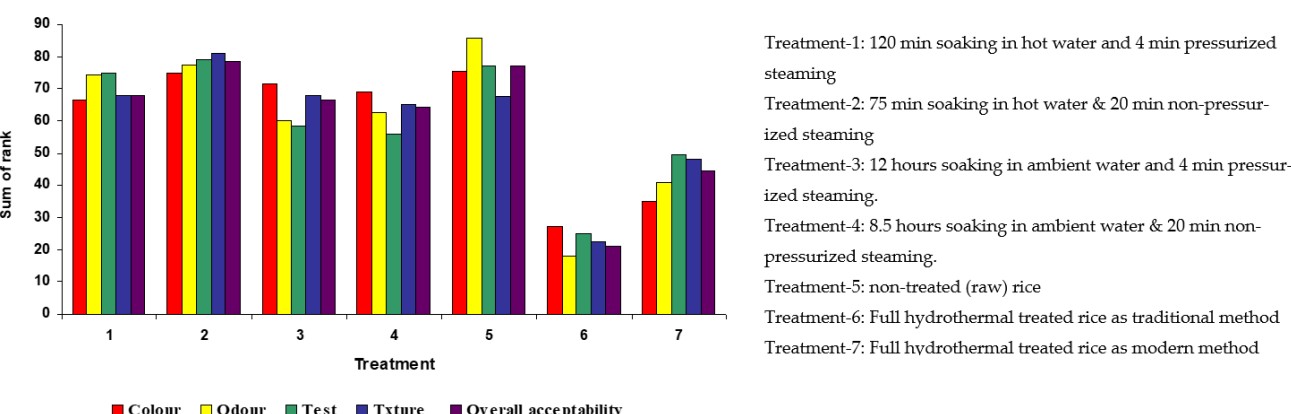

**Figure 7.** Sum of ranks of the organoleptic qualities of different hydrothermally treated rice.

The results clearly indicated that mild hydrothermal treatments significantly improved the colour of cooked rice in comparison to full hydrothermally treated rice. However, non-treated (raw rice) cooked rice samples showed the highest sum of ranks for basmati aroma. However, mild hydrothermally treated rice also showed higher sums of ranks. Full hydrothermally treated cooked rice reported the lowest sum of ranks. That revealed that mild hydrothermal treatments preserved basmati aroma because of the low duration of soaking and steaming. Accordingly, it can be concluded that mild hydrothermal treatment

improved the edible quality of basmati rice significantly. Out of four optimized mild hydrothermal treatments, the rice subjected to 75 min hot water soaking and 20 min non-pressurized steaming and soaking of the paddy for 120 min in hot water ($70 \pm 2\ °C$), and steaming the soaked paddy for 4 min by pressurized steam ($4\ kg/cm^2$) achieved similar organoleptic qualities as non-treated (raw) basmati rice.

### 3.7. Selection of the Optimized Mild Hydrothermal Treatment by DMRT beside Sensory Attributes

Table 4 shows the DMRT results of the sum of ranks for sensory attributes of the selected optimum mild hydrothermally treated rice, non-treated rice and full hydrothermally treated rice. See Figure 8 for microscopic images. It was clear from the results that mild hydrothermal treatment caused improvements in the organoleptic qualities of the rice. The mild hydrothermal treatment with 75 min hot water soaking and non-pressurized steaming for 20 min has been shown to achieve significant improvements of organoleptic qualities/palatability characteristics of rice. This treatment can produce rice similar to non-treated (raw) basmati rice. Hence, the rice subjected to 75 min hot water soaking and non-pressurized steaming for 20 min can be selected as the optimized mild hydrothermal treatment because it has shown all required qualities, such as colour, basmati aroma, taste, texture, high kernel whiteness, low kernel hardness, low grain breakage and high head rice yield percentage. Although it appeared similar to non-treated rice, it had better texture and taste than non-treated basmati rice. Moreover, 120 min hot water soaking and 4 min pressurized mild hydrothermal treatment can also be selected as the second-highest optimized treatment among four optimum mild hydrothermal treatments selected in the 1st phase of selection. Cooked rice texture was significantly improved by these mild hydrothermal treatments.

**Table 4.** DMRT results of the sum of ranks for sensory attributes of the selected optimum mild hydrothermally treated rice, non-treated rice and full hydrothermally treated rice.

| Treatment No. | Treatment | Colour | Odour | Taste | Texture | Overall Acceptability |
|---|---|---|---|---|---|---|
| 1 | 120 min soaking in hot water and 4 min steaming by pressurized steam | 66.5 [a] | 74.50 [a] | 75.00 [a] | 68.00 [b] | 68.00 [a] |
| 2 | 75 min soaking in hot water and 20 min non-pressurized steaming | 75.00 [a] | 77.50 [a] | 79.00 [a] | 81.00 [a] | 78.50 [a] |
| 3 | 12 h soaking in cold water and 4 min pressurized steaming | 71.50 [a] | 60.00 [b] | 58.50 [b] | 68.00 [b] | 66.50 [b] |
| 4 | 8.5 h soaking in ambient water and 20 min non-pressurized steaming | 69.00 [a] | 62.50 [a] | 56.00 [b] | 65.00 [b] | 64.50 [b] |
| 5 | Non-treated raw rice | 75.50 [a] | 85.50 [a] | 77.00 [a] | 67.50 [b] | 77.00 [a] |
| 6 | 45 h soaking in ambient water and 10 min pressurized steaming | 27.00 [b] | 18.00 [c] | 25.00 [c] | 22.50 [c] | 21.00 [c] |
| 7 | 4 h soaking in hot water and 10 min pressurized steaming | 35.00 [b] | 41.00 [b] | 49.50 [b] | 48.00 [c] | 44.50 [c] |

[a–c] Columns having same letter are not significantly different at $\alpha = 0.05$ by DMRT.

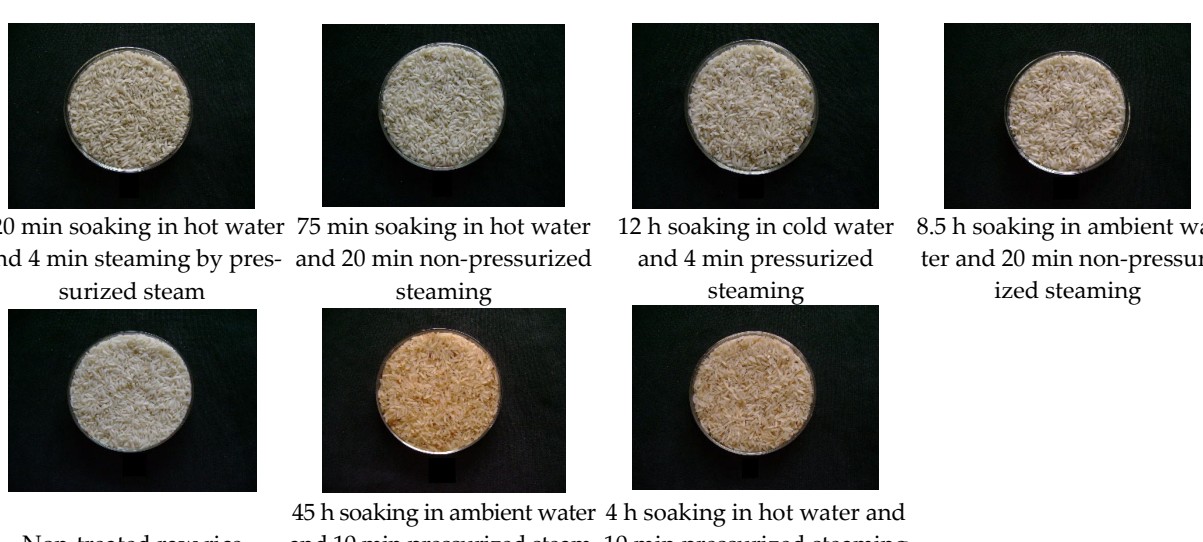

**Figure 8.** Different hydrothermally treated rice samples with non-treated rice.

## 4. Discussion

Low grain whiteness was observed in ambient-temperature water soaking and non-pressurized steam-treated samples. The short soaking and steaming times preserved the kernel whiteness. Gariboldi [10,11] and Gunathilake [5] explained that the change in of rice-kernel whiteness is probably due to gelatinization of the starch and disintegration of the protein bodies in the endosperm with increasing severity of the hydrothermal treatment of soaking and steaming. Miah et al. [12] observed that parboiling with increasing degrees of soaking and steaming, gradually increased the relative darkness. Full hydrothermally treated rice was observed to have high head rice yield. Head rice yield varies inversely with broken grain percentage; that is, low amounts of broken grain produce high head rice yield. Champa et al. [13] reported that reduction of grain breakage during milling operation and increasing of head rice yield is due to long soaking and steaming durations, which caused several physicochemical changes occurring in the rice caryopsis. Pillaiyar [4] reported that non-treated raw rice was soft and parboiled rice was hard, the hardness increasing with the degree of heat treatment such as steaming pressure and steaming duration during parboiling. Ali and Bhattacharya [14] reported that the high-pressure steaming brings about complete gelatinization of the grain up to the center, even at a low soaking moisture level.

It was observed that mild hydrothermal treatments preserved basmati aroma because of the low degree of heat treatment. Bhatt and Tomar [7] reported that numerous compounds are responsible for the aroma of aromatic rice. Aroma in basmati rice is formed by a blend of various volatiles. Full hydrothermal treatments remove cooked rice volatiles including free fatty acids, inactivate enzymes such as lipase and lipoxygenase, kill the embryo and decompose some antioxidants. It is also clear from the results that mild hydrothermal treatment improved the taste and texture of basmati rice in comparison full hydrothermally treated rice. Cooked rice texture can be significantly improved by mild hydrothermal treatments. Mohandoss et al. [15] reported that the hydrothermally treated rice eaters (consumers) would prefer moderately tender to slightly tough cooked rice and raw (non-treated) rice eaters would prefer tender cooked rice. Consumers may not prefer very tough rice. Other previous studies have also shown that a majority of consumers prefer whiter rice products Pillaiyar [4] and Chattopadhyay and Kunze [16] reported that the minimum level of hydrothermal treatments for Brazos variety was 270 min soaking at 65 °C and 5 min steaming at 103.51 kPa. Miah et al. [12] and Behra and Sutta [17] reported that a minimum hydrothermal treatment for long grain rice variety (BR4), of about 45-min soaking at 80 °C followed by steaming for about 10 min under 1 atmosphere excess pressure is necessary to improve all the required qualities of rice for better consumer preference.

## 5. Conclusions

It can be concluded that mild hydrothermal treatment consisting of 75 min hot water soaking and non-pressurized steaming for 20 min can preserve all required organoleptic qualities, basmati aroma, high kernel whiteness, low kernel hardness, low grain breakage and high head rice yield percentage. Although it appeared similar to non-treated rice, it had better texture and taste than non-treated basmati rice. Furthermore, 120 min hot water soaking and 4 min pressurized mild hydrothermal treatment can be selected as the second-highest optimized treatment among four optimum mild hydrothermal treatments selected in the 1st phase of selection. Finally, it can be concluded that mild hydrothermal treatment consisting of 75 min hot water soaking and non-pressurized steaming for 20 min is a suitable treatment to overcome the problems in basmati rice processing. However, industrial applications of this treatment need to be evaluated.

**Author Contributions:** D.M.C.C.G.: experimentation, analysis and writing, W.S. formatting and editing. All authors have read and agreed to the published version of the manuscript.

**Funding:** This manuscript received no external funding.

**Data Availability Statement:** Data is not available due to confidentiality.

**Conflicts of Interest:** The authors declare no conflict of interest.

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
