# Peer review of "Mild Hydrothermal Treatment for Improving Outturn of Basmati Rice"

_agriengineering, doi:10.3390/agriengineering5020062_

Round 1

Reviewer 1 Report

Dear Authors,

Please find the attached file below to see the comments.

Reviewer 2 Report

In this study authors examined influence of hydrothermal treatment to improvement of head rice yield. The study is scientifically interesting, but certain shortcomings should be corrected.

1. Please take into account the spelling of units, as well as typographical errors.

2. In the introduction part, it is necessary to add observations from recent studies that applied this technique to rice.

3. In order to more clearly monitor the impact of hydrothermal treatment on different parameters, show some of the results in Tables, also add standard deviations.

4. Compare the obtained results with literature.

5. Based on the obtained results, compare the hydrothermal method with other  methods applied for this purpose.

6. Does the hydrothermal treatment affect the mineral element contents? Have the authors considered this?

7. Figures 1-3 must be better.

8. How authors see the potential large-scale industrial application?

Author Response

Reviewer comments are addressed in the manuscript with track changes

Regards!

Champathi and Wijitha

Reviewer 3 Report

In this paper, authors provide a mild hydrothermal treatment method for Basmati rice soaking and steaming operations. Indeed, this is significative to reduce the level of grain breakage and increase head yield of rice during milling.

The overall impression left by the paper is very good. This paper can be published after minor modification. The main suggestions were listed below:

1. The references in the introduction of the paper are too old. It is recommended to appropriately cite papers published in the past three years.

2. The equipment and instruments used in the Materials and Methods section of the paper need to provide necessary information such as manufacturer, model, and accuracy.

3. The title of Table 3.1 is incorrect.

4. Authors should modify the citation methods, figures, tables, and reference formats according to the paper template

5. From the experimental data in the paper, we can see that the optimized method for parboiling of rice by the authors has a good effect on improving quality compared to traditional methods. But have there been any comparisons in terms of economy and practicality? Is there a breakthrough in cost savings and increased operational efficiency compared to traditional methods?

Thank you.

Author Response

Dear Reviewer, Please see the attachment. Thank you.

Regards!

Champathi and Wijitha

Round 2

Reviewer 1 Report

Dear Authors,

The authors have corrected minor linguistic errors, but have left major errors intact. They don't cut the Result and discussion part into two part as the template said.  Secondly, the major comments were answered in the cover letter, not in the article. So I suggest modifying the introduction part based on your response to comment_2. Still, the article contains only 16 references. There was no upgrade, just minor modification. But after the modifications, some references are still in not a required format. So please check the reference list again. For example, the title of the 4. citation is in italic. After the authors, it is necessary to use full stop, not semicolon (1., 2., 7., 8. and so on). The figures are still not self-explanatory. The minor problem is there is no dimension of the axis and explanation of the figure are in the text, not in the caption.  The major problem is, in fact, they contain non-existent combinations of treatments. For example, in the y-axis of the Figure1 shows 2.5 kg/cm2 pressure value, however it doesn't exist in material and methods (Table 1). How it is possible to visualize such data, if they did not exist?

So please, check the manuscript carefully again.

Reviewer 2 Report

Authors corrected manuscript, so norw it can be considered for publication.
